# Invasive and Non-Invasive Human Salmonellosis Cases Admitted between 2015 and 2021 in Four Suburban Hospitals in the Metropolitan Area of Milan (Italy): A Multi-Center Retrospective Study

**DOI:** 10.3390/pathogens12111298

**Published:** 2023-10-30

**Authors:** Gabriele Pagani, Marco Parenti, Marco Franzetti, Laura Pezzati, Francesco Bassani, Bianca Osnaghi, Laura Vismara, Claudia Pavia, Paola Mirri, Stefano Rusconi

**Affiliations:** 1UOC Malattie Infettive, ASST Ovest-Milanese, Ospedale Nuovo di Legnano, Via Papa Giovanni II s.n., Legnano, 20025 Milano, Italy; marco.franzetti@asst-ovestmi.it (M.F.); francesco.bassani@unimi.it (F.B.); stefano.rusconi@unimi.it (S.R.); 2Dipartimento di Scienze Biomediche e Cliniche “L. Sacco”, Università Statale degli Studi di Milano, Via G.B., Grassi 74, 20157 Milano, Italy; marco.parenti02@universitadipavia.it; 3UOC Microbiologia, ASST Ovest-Milanese, Ospedale Nuovo di Legnano, Via Papa Giovann II s.n., 20025 Legnano, Italyclaudia.pavia@asst-ovestmi.it (C.P.);

**Keywords:** *Salmonella*, non-typhoidal, invasive, NTS, zoonoses, northern-Italy

## Abstract

Non-Typhoidal Salmonellas (NTSs) are diffused worldwide. In Italy, more than 3500 cases are notified each year, but despite this, data about salmonellosis are scarce. Our multi-center, retrospective, descriptive study selected 252 patients with positive cultures for *Salmonella* spp. (feces 79.8%, blood 8.7%, feces and blood 7.5%), aiming to describe the epidemiological and clinical characteristics of this population. Bacteremic infections constituted 16.3% (41/252) of patients. Extreme ages (≤12 and ≥65 years old) accounted for 79.7%, but only elder ages were strongly associated with bacteremic infections (aOR 5.78). Invasive infections had a 7-fold higher mortality rate than non-invasive disease (9.8% vs. 1.4%, *p* = 0.013), with an overall rate of 2.8%. The two more represented serogroups were O:4 (52.8%) and O:9 (22.2%). The O:9 serogroup was strongly associated with a higher frequency of invasive infection (aOR 2.96, 95% CI 1.17–7.63). In particular, *S. napoli*, an emerging serovar in Europe, accounted for 31.7% of bacteremic infections and only 9.5% of non-bacteremic ones (*p* < 0.001). Antibiotic microbial resistance (AMR) observed a steep increasing trend and was detected in 60.4% of cases (122/202): amoxicillin/clavulanate, ampicillin, and gentamicin were most commonly involved (26.7%, 21.8%, 14.3%, respectively), while TMP/SMX, ciprofloxacin, and ceftriaxone were <10%. Resistance was, in general, less common in the O:9 serogroup, while amikacin resistance was significantly more common. Factors associated with an MDR Salmonella acquisition were time (OR 1.37 per year more), O:4 serogroup (OR 2.67), and being a woman (OR 2.11). The most frequent symptoms were diarrhea (90.5%) and fever (81.7%). In conclusion, our study highlights a high burden of NTS infections, leading to severe or fatal outcomes in frail patients. Furthermore, AMR shows an increasing trend with a concerning high prevalence of cephalosporine resistance compared to the rest of Europe.

## 1. Introduction

Salmonelloses are a group of oro-fecal food- and water-borne enteric or systemic infections sustained by Gram-negative bacteria belonging to the genus *Salmonella*. This genus includes two species, *S. enterica* and *S. bongori*, the former one being subdivided into six sub-species and more than 2600 serotypes, originally classified by Kaufman and White in subgroups based on differences between O, Vi, and H antigens [1].

The vast majority of human infections are caused by *S. enterica* subsp. *enterica*, the serotypes of which are clinically divided into invasive ones (Typhoidal Salmonellas, [TS]) and non-invasive ones (Non-Typhoidal Salmonellas [NTS]).

The former ones are classically represented by *S.* Typhi and *S.* Paratyphi, which are human-to-human transmitted without evidence of any animal reservoir, and they cause a severe and potentially fatal systemic illness known as Typhoid Fever (TF), which, albeit rare and usually imported in high-income countries (HIC), is still an important cause of disease and death in low-to-middle income countries (LTMIC), with an estimated 14,000,000 infections and 136,000 deaths worldwide during the year 2017 [2].

NTSs, on the other hand, are highly prevalent worldwide and tied to animal reservoirs, both domestic (pigs, turkeys, broilers, chickens, pet reptiles) and wild (hogs, wild birds), and represent one of the most frequent food-borne infection in both HIC and LMIC (the second most frequent in Europe, after Campylobacteriosis) [3]. While usually causing a self-limiting gastro-enteric syndrome, these can also cause an invasive disease similar to TF, more frequent in immunocompromised patients and both pediatric and geriatric patients, and tied to some NTS serotypes which are characterized by a higher invasive potential (invasive Non Typhoidal Salmonellas, iNTS), most commonly *S.* Typhimurium and *S.* Enteritidis [4,5]. Moreover, recent European studies found an increasing number of invasive infections caused by some serotypes of iNTSs [6].

Antimicrobial resistance (AMR) is also steadily increasing worldwide [7,8], including in Europe, where resistance is primarily connected to antimicrobial use in animal farming [9], and it is linked to higher rates of bloodstream infection and hospitalization [10].

In Italy, Salmonellosis is the most common food-borne infection, with more than 3500 notified cases per year [11]. In the last year, a surge of MDR strains was also described in our Country: a recent study set in Southern Italy in 2021 found a prevalence of MDR isolates in human and animal samples of 77% and 85% [12], with high levels of resistance to aminoglycosides, TMP/SMX, and fluoroquinolones, while another study on swine carcasses and sausages set in Central Italy found a 100% prevalence of aminoglycoside resistance, and high levels of tetracycline, penicillins (ampicillin and piperacillin), and TMP/SMX resistance [13]. Moreover, in recent years, there’s concern about the emergence of iNTS serotypes with high invasive potential, such as *S.* Enteritidis, *S.* Typhimurium, and *S.* Napoli. The latter is an emerging serovar in Europe, which recently sustained a multi-school outbreak in Milan (Northern Italy) in 2014, which was associated with elevated infection of bacteremia and hospitalization, even in young and healthy subjects [14].

While both TS and NTS are extensively studied in LMIC, especially South and South East Asia and Sub-Saharan Africa [4,5], differences in the epidemiological setting and study subjects’ characteristics render those findings difficult to apply in other geographical areas. On the contrary, studies on *Salmonella* infections in HIC (Europe and Italy, in particular) are scarce and usually limited to One-Health or veterinarian surveillance studies.

Our study aimed to describe the epidemiological and clinical characteristics of Salmonellosis cases admitted in four suburban hospitals in the outskirts of Milan’s metropolitan area, to analyze the differences in clinical and epidemiological characteristics between bacteremic and non-bacteremic infections, and to study the clinical and epidemiological factors associated with invasive disease (bacteremia) and the acquisition of drug-resistant strains in the hospitalized population.

## 2. Materials and Methods

### 2.1. Study Setting

This is a multicenter, retrospective, descriptive study carried out at ASST (Azienda Socio Sanitaria Territoriale) Ovest Milanese, a sanitary “hospital agency” (Azienda Ospedaliera) comprising four hospitals (Legnano, Cuggiono, Magenta, and Abbiategrasso) spread in the western part of the metropolitan area of Milan, Italy.

### 2.2. Study Procedures

We retrospectively selected for our study all patients with a positive culture for *Salmonella* spp. on at least one sample among stools, blood, urine, or other bodily fluids. Anagraphic, epidemiological, and clinical data were retrieved using the hospital’s informatics systems.

For every patient, we recorded:Anagraphic data (name, age, sex, nationality, place of residence).Date and hospital of diagnosis.Body fluid on which *Salmonella* grew (defined as blood, feces, urine, or other).Species, serogroup, and serotype (defined as the O-antigen group and serotype’s name, when available) of the isolate.AMR pattern of the isolate.Clinical history (including a history of immunodepression).Area of hospitalization (defined as OBGYN/Pediatrics, Internal Medicine Division, Surgery Division, Intensive Care, Emergency Department).Symptoms at presentation.Date of symptoms’ onset.Blood tests at presentation.Date and length of hospital admission.Antimicrobial treatment (including empirical and antibiogram-aimed treatment).Infectious Diseases Specialist consultation during the hospital stay.Outcome (defined as dead or discharged).

### 2.3. Definitions

*Salmonella* infection was defined as the growth of *Salmonella* spp. on any material. Bacteriemia (Invasive disease) was defined as positive blood cultures (BCx) for *Salmonella* spp., both alone or associated with positive cultures on any other media.

Every patient in the study underwent at least one BCx either at presentation or during the hospital stay.

Immunodepression status, if present, was further categorized as iatrogenic (if caused by any immunosuppressive therapy), onco-hematological (if primarily associated with onco-hematological malignancies and their treatment), and HIV-related (if associated with HIV infection, regardless of CD4+ lymphocyte count).

### 2.4. Microbiology

The sowing of fecal samples for Coproculture (for *Salmonella*, *Shigella*, and *Campylobacter* research) was performed in total automation with the WASPLab System. Samples were sown on solid medium McC (MacConkey agar), in Selenite liquid medium, and incubated for 24 h in a thermostat at 37 °C. The liquid medium was then subcultured on DCA (Desoxicholate Citrate Lactose Agar) plates and incubated aerobically at 37 °C for 24 h.

Bacterial growth was assessed by the type of colonies present on the plates.

*Salmonella* strains are thus typed with Wellcolex Color *Salmonella*, which provides a simple and rapid qualitative latex test procedure for the screening, detection, and presumptive identification of the serogroup in a reagent.

Serotyping of *Salmonella* strains was performed by Somatic (0) and Flagellar (H) phases determination, then using specific antigens’ monovalent sera.

The species identification was performed using Maldi-TOF (Matrix-Assisted Laser Desorption/Ionization—Time Of Flight) mass spectrophotometry. Antibiograms were obtained using the VITEK system (microdilution in plate, BioMerieux). The Minimum Inhibitory Concentration (MIC) for each antibiotic was interpreted according to EUCAST’s (European Committee on Antimicrobial Susceptibility Testing) most recent guidelines (rev 13 2023).

Isolated strains are stored at −20 °C and sent to the Lombardy Region ATS (Health Protection Agencies) Reference Center for epidemiological investigation and genotypic completion of the species.

### 2.5. Statistical Analyses

Continuous variables were described as mean and standard deviation or median and inter-quartile range (IQR) and confronted using standard parametric and non-parametric tests as appropriate. Discrete variables were described as absolute and relative frequencies and confronted using χ^2^ and Fisher’s Exact Test as appropriate.

Categorical data were analyzed using linear-by-linear associations for trends over time periods.

The risk of invasive disease (bacteremia) and the risk of drug-resistant strain detection was evaluated using a logistic regression model, also correcting for gender, age, the time between symptoms’ onset and diagnosis, serogroup (O:4, O:9, or other serogroups) and year of diagnosis.

Data were anonymized and gathered on a Microsoft Excel Spreadsheet; data analysis was performed using the IBM SPSS Statistics version 25.

*p*-values < 0.05 were considered significant for two-tailed tests.

## 3. Results

### 3.1. Demographics and Hospital Admission

A total of 252 patients were enrolled in our study; invasive salmonellosis accounted for 16.3% (41/252) infections. See Figure 1 for the number of cases during each year of our study.

M:F ratio was 1.45:1. Median age was 15 years (IQR 4–71), with a dramatic difference between bacteremic and non-bacteremic infections (68 vs. 10 years). Young patients ≤ 12 years old represented 48.4% of patients (*n* = 122), while almost one-third (*n* = 79; 31.3%) were ≥65 years of age (see Figure 2 for age and sex distribution).

The majority of patients (*n* = 132, 52.4%) were admitted to the pediatric/OB-GYN department, 76 (30.1%) patients were admitted to the internal medicine department, 34 (13.5%) underwent observation in the ED and were discharged home without a further hospital stay, 8 (3.2%) were admitted in the Surgery department, and 2 (0.8%) were admitted to the ICU. Bacteremic patients were more often admitted to the Internal Medicine department (41.6%), while the pediatric/OB-GYN department accounted for most of the non-bacteremic infections (57.8%).

Information about the length of hospital stay was available in 90.6% of patients (*n* = 232), with a median hospital stay of 5 days (IQR 3–9), with a significantly longer stay recorded for bacteremic patients (10 vs. 5 days, *p* < 0.001). The interval between symptoms onset and hospitalization was 3 days (IQR 3–9).

### 3.2. Site of Salmonella spp. Isolation

Bacterial growth was obtained on feces in 79.8% of cases, blood in 8.7%, and both blood and feces in 7.5%. Positive urine cultures accounted for 1.5% of cases, while in 2.5% of cases, the diagnosis was made on other bodily fluids (pleural fluid, synovial fluid, peritoneal fluid, sperm). All admitted patients underwent at least one set of blood cultures.

### 3.3. Mortality

A total of 7 patients (2.8%) died during the hospital stay. Invasive infections had a 7-fold higher mortality rate concerning non-invasive disease (9.8% vs. 1.4%, *p* = 0.013).

See Table 1 for baseline demographic and epidemiological characteristics of NTS vs. iNTS.

### 3.4. Serogroups and Serotypes Prevalence

Isolated serogroups and serovar are listed in Table 2. More than half of the infectious were sustained by the O:4 serogroup (*n* = 133, 52.8%), most frequently represented by *S.* Typhimurium (98/252, 38.1%). The O:9 group was the second most frequent, with 56 infections (22.2%), primarily sustained by *S.* Napoli (33/252, 15.5%) and *S.* Enteritidis (18/252, 7.1%).

In Table 3, we compared the frequencies of serogroups according to invasive or non-invasive disease; a higher frequency of O:9 infections in the bacteremic vs. non-bacteremic group was recorded (43.9% vs. 18%, *p* < 0.001), while O:4 was more frequent in the non-invasive compared to the invasive infection group (57.7% vs. 26.8%, *p* = 0.001). This is also confirmed using the same analysis performed on the most frequently isolated serotypes (Table 4), where *S.* Typhimurium was isolated less frequently in bacteremic vs. non-bacteremic patients (17 vs. 43.1%, *p* = 0.002) while bacteremic infections were sustained by *S.* Napoli in 31.7% of cases compared to 9.5% in non-bacteremic subjects (*p* < 0.001).

### 3.5. Antimicrobial Treatments and Antimicrobial Resistance

Antibiograms were available for 202 patients (80.2%), and at least one antibiotic resistance was detected in 60.4% of cases (122/202). Resistance to amoxicillin/clavulanate, ampicillin, and gentamicin were the most common in our study (21.4%, 16.7%, and 11.5%, respectively, see Table 5), while resistance to TMP/SMX (7.5%), ciprofloxacin (7.5%) and ceftriaxone (4%) were less common. No resistance to carbapenems was found in our study. Approximately one-fourth (50/202; 24.8%) of the isolates were resistant to 2 or more classes of antibiotics.

The frequency of resistant isolates did not significantly differ between the bacteremic and non-bacteremic groups (at least one resistance: 59.4% vs. 60.6%, *p* = 0.999; resistance to two or more classes of antibiotics: 28.1% vs. 24.1%, *p* = 0.83).

Absolute and relative AMRs for each year of our study are reported in Figure 3. There were no recorded resistances in 2015, and there was a significant increasing trend between the first and last year of our study (44.4% vs. 87.1%, *p* = 0.001).

The frequency of serogroups significantly differs between subjects that presented a drug-resistant isolate compared to fully sensible isolates; in particular, the O:4 serogroup was more frequent in the former group (63.9% vs. 36.3%, *p* < 0.001), while the O:9 group Salmonellas were more frequent in the latter (8.2% vs. 37.5%, *p* < 0.001), see Appendix A. It is interesting to notice that, while resistance, in general, was less common in the O:9 group Salmonellas, amikacin resistance was significantly more common in the O:9 group compared to the others (14.3% vs. 6.6%, *p* < 0.034) and the same trend, albeit not significant, is seen for gentamycin (16.1% vs. 9.7%, *p* = 0.068), suggesting aminoglycoside-class resistance. Resistance to amoxicillin/clavulanate, ampicillin, and TMP/SMX, on the contrary, was almost not present in the O:9 group compared to all other serogroups (respectively: 26% vs. 1.8%, *p* < 0.001; 21.4% vs. 0%, *p* < 0.001; 9.2% vs. 0%, *p* = 0.027; see Appendix A).

Antimicrobial treatment was started in little less than half of our study subjects (*n* = 120, 47.6%), with a much higher frequency in bacteremic patients (92.7%) vs. non-bacteremic (38.9%, *p* < 0.001).

See Figure 3 for absolute and relative frequencies of the chosen first-line antibiotic agents.

Susceptibility testing was available for 96/120 antibiotic-treated patients, according to which the chosen regimen was inadequate in 16 cases (16.7%).

AMK amikacin, AMC amoxicillin/clavulanic acid, AMP ampicillin, AMP + SUL ampicillin + sulfamethoxazole, CAZ ceftazidime, CAF ceftriaxone, CIP ciprofloxacin, CLR clarythromicin, IPM imipenem, LVX levofloxacin, MEM meropenem, MET metronidazole, TZP piperacillin/tazobactam, and VAN vancomicin.

### 3.6. Signs and Symptoms

The most frequent signs and symptoms at presentation in invasive and non-invasive infections are reported in Table 6.

Diarrhea was the absolute most frequent sign (90.5%), with a slightly lower frequency in invasive infections (78% vs. 92.9%, *p* = 0.007). Fever, vomiting, abdominal pain, and malaise were the next most represented four signs and symptoms (81.7%, 39.7%, 31%, and 11.5%, respectively), without significant differences between the two groups, except for a higher frequency of malaise in the bacteremic group (24.4% vs. 9%, *p* = 0.008).

### 3.7. Laboratory Values

Median laboratory values and their differences between bacteremic and non-bacteremic patients are reported in Table 7. It’s interesting to notice that apart from the PCR value, all the other median values are within the limits of normality.

Bacteriemic patients presented a significantly lower median platelet count (179,000 vs. 236,000/mmc, *p* < 0.001) and WBC count (6600 vs. 9300/mmc, *p* < 0.001) while having significantly higher median values of glucose (119 vs. 106, *p* = 0.008), ALT transaminases (27 vs. 20 U/L, *p* = 0.002) and creatinine (0.96 vs. 0.56 g/dL, *p* = 0.001).

### 3.8. Factors Associated with Bacteremia

Table 8 describes the factors associated with an invasive infection in univariate and multivariate analyses. In the former, an O:4 serogroup sustained infection was associated with a lower frequency of invasive infection (OR 0.39, 95% CI 0.17–0.89, *p* = 0.025) while being aged 12–65 or >65 years old was associated with invasive infection (OR 12–65 yo vs. <12 yo: 3.21, 95% CI 1.24–8.29, *p* = 0.016; OR >65 yo vs. <12 yo 4.79, 95% CI 2.06–11.16, *p* < 0.001).

In multivariable analyses, however, the protective role of O:4 serogroup lost significance, while O:9 serogroup infections were strongly associated with bacteremia (aOR 2.96, 95% CI 1.15–7.63, *p* = 0.025); age continued to be significantly associated with an invasive infection (aOR 12–65 yo vs. <12 yo: 3.35, 95% CI 1.17–9.56, *p* = 0.024; aOR > 65 yo vs. <12 yo 5.78, 95% CI 2.28–14.67, *p* < 0.001).

### 3.9. Factors Associated with Antibiotic-Resistant Salmonella Infection

Factors associated with acquiring AMR *Salmonella* infections in univariate and multivariate analyses are presented in Table 9. In univariate analyses, factors associated with an MDR *Salmonella* acquisition were acquiring the disease later during our study (OR 1.37 per every year more, 95% CI 1.13–1.67, *p* = 0.002), being infected with an O:4 serogroup *Salmonella* (OR 2.67, 95% CI 1.39–5.14, *p* = 0.003) and being a woman (OR 2.11, 95% CI 1.17–3.84, *p* = 0.014).

Being infected by an O:9 serogroup *Salmonella* (OR 0.41, 95% CI 0.17–0.98, *p* = 0.045) and being aged 12–65 (OR 12–65 yo vs. <12 yo: 0.48, 95% CI 0.23–0.99, *p* = 0.045) were inversely associated with an MDR infection.

The same associations were confirmed in multivariate analyses: an O:4 infection (aOR 2.18, 95% CI 1.06–4.47, *p* = 0.035) and being a woman (aOR 2.94, 95% CI 1.46–5.87, *p* = 0.002) was associated with drug-resistant salmonellosis, while an O:9 infection (aOR 0.22, 95% CI 0.08–0.63, *p* = 0.004) and being aged 12–65 (aOR 0.39, 95% CI 0.162–0.914, *p* = 0.031) were inversely associated with the outcome.

## 4. Discussion

Italy is disproportionately affected by salmonellosis, with a more than doubled incidence compared to the European average [15]. Despite this and an efficient veterinary surveillance program, clinical studies about Salmonelloses and iNTS are scarce.

In our study, we found a total of 252 patients admitted to our hospital for salmonellosis. Among these, 41 had evidence of bloodstream infection. Only two cases, however, were caused by TS (one case sustained by *S.* Typhi, one by *S.* Paratyphi A), both imported.

Extreme ages were mostly affected, with almost half of the infections found in patients ≤ 12 years (*n* = 122; 48.4%) and circa one-third (*n* = 79; 31.3%) in subjects ≥ 65 years, which is consistent with data from the EnterNET network [16]. Interestingly, considering only the 41 patients with bloodstream infection, we find a similar prevalence of age groups, with 50% of patients being ≥65 and 23% ≤12 years old.

Immune impairment, on the other hand, was rare. This is a very different scenario from Asian and African studies (focused mainly on iNTS), where immune impairment, especially HIV-related, was highly prevalent, and the median age was dramatically lower [4,5]. This dramatic difference is mainly due to demographic differences in the cohort, but it does also suggest that an overt immune-system impairment is not a necessary factor to develop an invasive infection, especially in the extreme age groups, as confirmed by a large recent outbreak in children in several schools in Milan [14].

Intra-hospital mortality was low overall (2.8%), but it is worth noticing that considering only bacteremic patients, the case fatality ratio approached 10%, a value confirmed by a recent global meta-analysis of iNTS, which found a 9.9% CFR in Europe as a whole and an 8.8% CFR considering only western Europe, which is a notably high burden of mortality, especially considering that NTSs are usually considered as a relatively harmless disease. As a form of comparison, an identical CFR was found for Invasive Pneumococcal Disease in Sweden in 2017 [17].

Regarding isolated serogroups (Table 2), the majority (52.8%) of *Salmonella* infections in our study were sustained by O:4 group (comprising *S.* Typhimurium, Typhimurium var. monofasica and *S.* Brandenburg) followed by O:9 group (comprising *S. napoli* and *S. enteritidis*) and O:7 group (*S.* Infantis and *S.* Brandaerup). Two-thirds (65.8%) of infections were sustained by four serotypes: *S.* Typhimurium, *S.* Napoli, *S.* Enteritidis, and *S.* Infantis (38.1%, 15.5%, 7.1%, and 5.1%, respectively). This is partially in agreement with recent European surveillance data, where Enteritidis, Typhimurium, and Typhimurium var. monofasica represented the majority (74.8%) of infections in humans and are among the most frequently isolated serotypes in all domestic animal reservoirs [3]. *S.* Napoli, on the other hand, is rarely isolated from domestic animal reservoirs, as it’s probably linked to environmental reservoirs and has seen a steady increase in incidence in recent years, both in Europe and in Italy [18,19]. In Northern Italy, it recently caused a multi-school outbreak characterized by elevated rates of bacteremia and hospitalization [14], and it was the first cause of iNTS in Lombardy between 2011 and 2014 [20].

Anti-microbial resistance (AMR) was highly prevalent in our study, with more than half of the isolates resistant to at least one molecule and an increasing trend, leading to almost 90% of isolates being resistant to at least one antibiotic in 2021 (as opposed to less than half in 2016). Almost 25% of the isolates were resistant to more than one class of antibiotics.

It must be noted, however, that our study is based on hospitalized patients, and it is known that drug resistance is associated with more severe disease and hospitalization [10], leading to possible selection bias.

Penicillins (both alone and in combination with a beta-lactamase inhibitor) and aminoglycoside were the most common resistances, followed by fluoroquinolones and cephalosporines; carbapenem resistance was not found in our study (see Table 5). This is in partial agreement with European data on both human and animal isolates, where complete susceptibility was observed in about half of the cases, and resistance to ampicillin and sulfonamides were the more prevalent, possibly due to antimicrobial use in farming (tetracyclines were not considered in our study because they’re not part of EUCAST susceptibility testing in human salmonellosis). Reported resistance to third-generation cephalosporins is very low or non-existent in Europe; in Italy, however, the prevalence of resistance to cefotaxime and ceftazidime in both animals and humans, is more frequent compared to the rest of Europe, as confirmed by our data, where a low, albeit significant prevalence of resistance to 3rd generation cephalosporines is present [21].

Resistance to aminoglycosides was frequent in our study but not usually reported by surveillance studies. This could be due to the fact that gentamicin and amikacin resistance were more common in the 0:9 group Salmonellas, particularly in the Napoli serovar, which is not usually sampled in surveillance studies, both in animals and men, as this serovar is mainly tied to wastewaters and wild animal reservoirs [18].

Clinical presentations of salmonellosis were not significantly different between bacteremic and non-bacteremic patients, a part for a higher prevalence of reported malaise. This suggests caution as the initial presentation usually does not help in discriminating invasive infections.

Bacteriemic patients, on the other hand, usually had worse median laboratory values: in particular, a lower WBC and PLT count and higher glucose, ALT, and creatinine levels, suggesting a more profound systemic involvement. It is interesting to notice, however, that median laboratory values at presentation (except for PCR), both in invasive and non-invasive infections, were not outside of the reference range, suggesting again caution in using laboratory values to discriminate severe infections. These findings are consistent with the Vietnamese study on iNTSs despite being based only on invasive infections and a much higher prevalence of immunocompromised patients [5].

To further help discriminate which patients are at risk for invasive infections, we studied the factors associated with developing an invasive infection: being aged > 65 and 12–65 are both associated with invasive infections compared to children (with a higher association with elderly patients), as it is being infected with an O:9 serogroup salmonella. The latter association could be explained by the known invasive potential of both *S.* Napoli [20] and *S.* Enteritidis [5], which together represented 91.1% of isolated O:9 serovars in our study, suggesting particular caution in elderly patients infected with these serotypes.

We also studied the factors associated with AMR *Salmonella* (at least one resistance): as already described before, being infected later in our study and O:9-group *Salmonella* infection are associated with resistance, while O:4 infection is protective against infection with resistant bacteria. Interestingly, the female sex was strongly and independently associated with acquiring an antimicrobial-resistant infection: while this could be a spurious correlation due to missing factors in the analysis, the role of women traditionally doing most of the cooking in Italian households could expose them to a higher risk of infection from raw meat and eggs from antibiotic exposed farmed animals.

### Limitations

Our study is retrospective; this could lead to reporting errors and missing or inaccurate data, especially regarding the clinical history and clinical presentation.

Moreover, our study is based only on hospitalized patients in a restricted geographical area, which leads to the selection of more severe infections and warrants caution in interpreting and generalizing some results, such as the prevalence of serogroups and resistant isolates and CFR.

## 5. Conclusions

In conclusion, our study suggests a higher-than-expected burden of iNTS infections, which are still poorly studied in both Italy and HIC in general and could represent an important cause of morbidity and mortality in some categories of patients, such as the elderly.

Antimicrobial resistance was highly prevalent in our study, with a steadily increasing trend during the 5 years considered in our study, with a higher prevalence of cephalosporines resistance compared to the rest of Europe and an almost universal resistance to penicillins in some serogroups such as O:4, possibly linked to poorly controlled antibiotic use in both human and animal health and suggesting an urgent one-health approach to revert this worrying trend.

## Figures and Tables

**Figure 1 pathogens-12-01298-f001:**
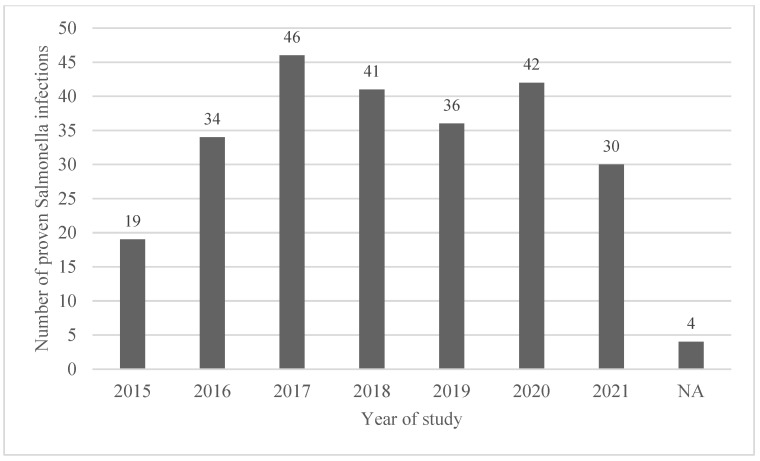
Number of proven *Salmonella* infections in hospitalized patients by year of observation (2015–2021).

**Figure 2 pathogens-12-01298-f002:**
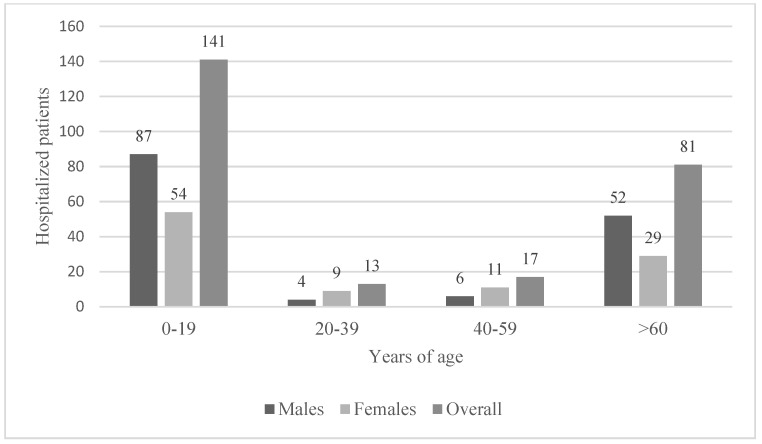
Hospitalized patients: distribution by age and sex.

**Figure 3 pathogens-12-01298-f003:**
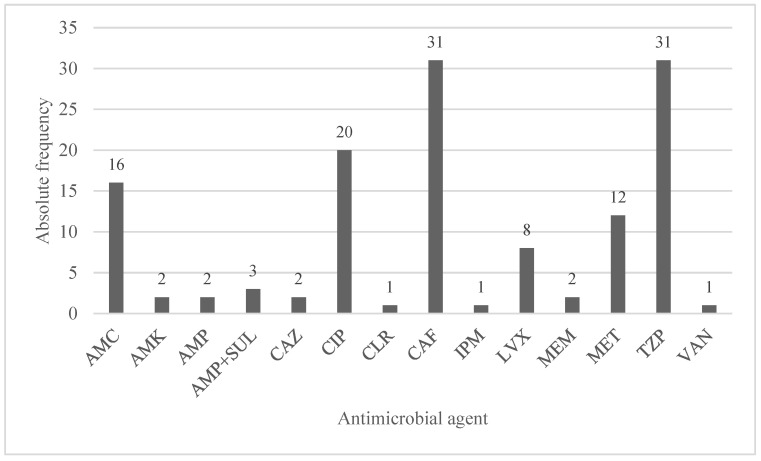
Absolute frequencies of the chosen first-line antibiotic agents.

**Table 1 pathogens-12-01298-t001:** Characteristics of the study population: patients with invasive (bacteriemic) and non-invasive (non-bacteriemic) *Salmonella* infection, and overall.

	Bacteriemic Infections	Non-Bacteriemic Infections	Overall	*p*-Value
Proven cases, N (%)	41 (16.3%)	211 (83.7%)	252	
Age (median, IQR)	68 (19–81)	10 (3–67)	15 (4–71)	<0.001
Sex, N (%)				0.260
Male	21 (51.2)	128 (60.7)	149 (59.1)	
Female	20 (48.8)	83 (39.3)	103 (40.9)	
Male/Female	1.05:1	1.51:1	1.45:1	

Hospital unit, N (%)				0.001
OB/GYN and Pediatrics	10 (24.5)	122 (57.8)	132 (52.4)	
Internal Medicine	17 (41.6)	59 (28)	76 (30.2)	
Emergency Room	12 (29.4)	22 (10.5)	34 (13.4)	
Surgery	2 (4.5)	6 (2.8)	8 (3.2)	
Intensive Care	0	2 (0.9)	2 (0.8)	

Municipality of origin, N (%)				0.172
Abbiategrasso	5 (12.2)	28 (13.4)	33 (13.1)	
Magenta	5 (12.2)	64 (30.3)	69 (27.4)	
Cuggiono	7 (17.1)	30 (14.3)	37 (14.4)	
Legnano	14 (34.1)	52 (24.6)	66 (26.2)	
Other	10 (24.4)	37 (17.5)	47 (18.6)	

Immune-suppression, N (%)				0.078
Iatrogenic	4 (9.8)	2 (0.9)	6 (2.3)	
Oncohaematological	1 (2.4)	5 (2.4)	6 (2.3)	
HIV-related	0	2 (0.9)	2 (0.8)	

Length of hospital stay, days (median, IQR) *	10 (8–13)	4 (3–7)	5 (3–9)	<0.001
Symptoms-to-diagnosis interval (median, IQR)	4 (1–6)	3 (2–5)	3 (2–5)	0.055
ID consultation N (%)	6	21	27	0.263

Outcome, N (%)				0.013
Death	4 (9.8%)	3 (1.4%)	7 (2.8%)	
Discharge	37 (90.2%)	208 (98.6%)	245 (97.2%)	

Antibiotic regimen, N (%)	38 (92.7%)	82 (38.9%)	120 (47.6%)	<0.001
Inappropriate first-line regimen, N (%)	2 (5.3%)	14 (17.5%)	16 (13.3%)	0.135

* data available for 232/252 patients (90.6%).

**Table 2 pathogens-12-01298-t002:** Distribution of isolated serogroups and serovars in hospitalized patients. ND = not determined.

Serogroups/Serovars	Absolute Frequency	Frequency %
O:4	133	52.8
Typhimurium	98	38.1
Brandenburg	9	3.5
Derby	9	3.5
Var. Monofasica	8	3.1
Agbeni	1	0.4
Bredeney	1	0.4
Chester	1	0.4
Kapemba	1	0.4
Saintpaul	1	0.4
ND	10	4.0
O:9	56	22.2
Napoli	33	15.5
Enteritidis	18	7.1
Typhi	2	0.6
Panama	1	0.4
ND	4	1.6
O:7	25	9.9
Infantis	13	5.1
Braenderup	1	0.4
Isangi	1	0.4
Livingstone	1	0.4
Rissen	1	0.4
Strathcona	1	0.4
Thompson	1	0.4
Virchow	1	0.4
ND	4	1.6
O:8	9	3.6
Bovismorbificans	4	1.6
Goldcoast	2	0.8
Blockley	1	0.4
ND	2	0.8
O:3, 10	2	0.8
Muenster	1	0.4
London	1	0.4
O:11	2	0.8
Veneziana	2	0.8
O:13	2	0.8
Kedougou	1	0.4
Poona	1	0.4
O:2	1	0.4
Paratyphi A	1	0.4
ND	14	5.6
Overall	252	100

**Table 3 pathogens-12-01298-t003:** Differences in frequencies of isolated serogroups in bacteriemic and non-bacteriemic infections (hospitalized patients).

Serogroups, N (%)	Bacteriemic Infections (*n* = 41)	Non-Bacteriemic Infections (*n* = 211)	Overall (*n* = 252)	*p*-Value
O:4	11 (26.8)	122 (57.7)	133 (52.8)	0.001
O:9	18 (43.9)	38 (18)	56 (22.2)	<0.001
O:7	5 (12.3)	20 (9.5)	25 (9.9)	0.594
ND	6 (14.6)	16 (7.5)	22 (8.7)	/
O:8	0	9 (4.3)	9 (3.6)	/
O:3, 10	0	2 (1.0)	2(0.8)	/
O:11	0	2 (1.0)	2 (0.8)	/
O:13	0	2 (1.0)	2 (0.8)	/
O:2	1 (2.4)	0	1 (0.4)	0.163
Overall	41 (100)	211 (100)	252 (100)	

**Table 4 pathogens-12-01298-t004:** Differences in absolute frequencies of the topmost isolated serovars in bacteriemic and non-bacteriemic infections (hospitalized patients).

Serovars, N (%)	Bacteriemic Infections (*n* = 41)	Non-bacteriemic Infections (*n* = 211)	Overall (*n* = 252)	*p*-Value
Typhimurium	7 (17)	91 (43.1)	98 (38.9)	0.002
Napoli	13 (31.7)	20 (9.5)	33 (13.1)	<0.001
Enteritidis	5 (12.2)	13 (6.2)	18 (7.1)	0.332
Infantis	2 (4.9)	11 (5.2)	13 (5.6)	0.929
Derby	0	9 (4.3)	9 (3.6)	0.362
Brandenburg	2 (4.9)	7 (3.3)	9 (3.6)	0.361
Other	12 (29.3)	60 (28.4)	72 (28.6)	

**Table 5 pathogens-12-01298-t005:** Overview of antimicrobial resistance according to the available antibiograms in hospitalized patients.

	Frequency of Antimicrobial Resistance, N (%) *
Isolates showing resistance to at least 1 antibiotic	122 (60.4)
Amikacin	22 (10.9)
Gentamicin	29 (14.3)
Amoxicillin/clavulanate	54 (26.7)
Ampicillin	44 (21.8)
Cefepime	1 (0.5)
Cefotaxime	4 (2.0)
Ceftazidime	1 (0.5)
Ceftriaxone	10 (5.0)
Ciprofloxacin	19 (9.4)
Fosfomycin	0
Meropenem	0
Piperacillin	26 (12.9)
Tigecyclin	12 (5.9)
Piperacillin/tazobactam	4 (2.0)
Trimethoprim/sulfamethoxazole	18 (8.9)
Ertapenem	0

* data available for 202 patients.

**Table 6 pathogens-12-01298-t006:** Most frequently reported signs and symptoms in hospitalized patients with and without bacteriemic infections.

Signs and Symptoms, N (%)	Bacteriemic Infections (*n* = 41)	Non Bacteriemic Infections (*n* = 211)	Overall (*n* = 252)	*p*-Value
Diarrhea	32 (78.0)	196 (92.9)	228 (90.5)	0.007
Fever	35 (85.4)	171 (81.1)	206 (81.7)	0.292
Vomit	16 (39.0)	84 (39.8)	100 (39.7)	0.535
Abdominal pain	14 (34.1)	64 (30.3)	78 (31)	0.377
Malaise	10 (24.4)	19 (9.0)	29 (11.5)	0.008
Nausea	3 (7.3)	12 (5.7)	15 (6)	0.435
Sensory alterations	4 (9.8)	11 (5.2)	15 (6)	0.213
Dyspnea	5 (12.2)	8 (3.8)	13 (5.2)	0.042
Neurological symptoms	8 (19.5)	2 (0.9)	9 (3.6)	0.505
Pneumonia	1 (2.4)	1 (0.5)	2 (0.8)	0.299
Septic arthritis	0	1 (0.5)	1 (0.4)	0.842

**Table 7 pathogens-12-01298-t007:** Median laboratory values and Interquartile Ranges (IQR) differences in hospitalized patients with and without bacteriemic infections.

	Bacteriemic Infections (*n* = 41) Median (IQR)	Non Bacteriemic Infections (*n* = 211) Median (IQR)	Overall (*n* = 252) Median (IQR)	*p*-Value
Hb (g/dL)	13.7 (12.1–14.2)	13.2 (12.1–14.1)	13.2 (12.1–14.1)	0.354
Hct (%)	39.2 (35.0–43.8)	39.2 (36.5–42.3)	39.4% (36.4–42.9)	0.863
PLT (/mcL)	179,500 (123,750–240,500)	236,000 (182,500–299,000)	239,000 (175,000–287,000)	<0.001
WBC (/mcL)	6600 (4375–9428)	9300 (7500–13,200)	9100 (6700–12,450)	<0.001
N (/mcL)	5150 (3019–7591)	6836 (4357–9728)	6396 (4230–9379)	0.635
L (/mcL)	943 (496–1441)	1124 (770–2165)	1189 (746–2081)	0.970
M (/mcL)	617 (284–891)	831 (607–1109)	807 (555–1091)	0.395
E (/mcL)	6 (0–15)	9 (0–28)	8 (0–27)	0.629
B (/mcL)	20 (10–36)	28 (17–45)	27 (16–45)	0.912
LDH (U/L)	258 (178–405)	242 (202–310)	244 (200–331)	0.270
AST (U/L)	36 (23–53)	30 (21–40)	30 (21–41)	0.093
ALT (U/L)	27 (21–43)	20 (16–29)	22 (17–31)	0.002
BR (mg/dL)	0.63 (0.4–1.01)	0.56 (0.37–0.77)	0.58 (0.37–0.79)	0.185
Glu (mg/dL)	119 (98–143)	106 (89–122)	107 (90–126)	0.008
BUN (mg/dL)	35 (23–74)	26 (19–50)	27 (19–57)	0.036
sCr (mg/dL)	0.96 (0.79–1.78)	0.56 (0.38–1.15)	0.71 (0.39–1.22)	0.001
CRP (mg/L)	6.90 (3.20–14.6)	6.12 (2.68–10.83)	6.20 (2.79–11.40)	0.314
PT (INR)	1.17 (1.09–1.34)	1.23 (1.09–1.45)	1.22 (1.09–1.42)	0.421
aPTT	0.97 (0.91–1.07)	0.98 (0.90–1.08)	0.9 (0.9–1.07)	0.681
Na^+^ (mEq/L)	137 (134–140)	136 (133–138)	136 (133–139)	0.359
K^+^ (mEq/L)	3.78 (3.4–4.18)	3.93 (3.53–4.20)	3.9 (3.5–4.2)	0.253

Hb: hemoglobin; Hct: hematocrit; PLT: platelets; WBC: white blood cells; N: neutrophils; L: lymphocytes; M: monocytes; E: eosinophils; B: basophils; LDH: lactate dehydrogenase; AST: aspartate aminotransferase; ALT: alanine aminotransferase; BR: bilirubin; Glu: blood glucose; BUN: blood urea nitrogen; sCr: serum creatinine; CRP: C reactive protein; PT: prothrombin time; INR: international normalized ratio; aPTT: activated partial prothrombin time; Na^+^: sodium; K^+^: potassium; g/dL: grams/deciliter; mcL: microliter; U/L: Unit/Liter; mEq/L: milliequivalent/Liter.

**Table 8 pathogens-12-01298-t008:** Factors associated with bacteremic infection: univariate and multivariate analysis.

	Univariate			Multivariate		
	aOR	95% CI	*p*-Value	aOR	95% CI	*p*-Value
Age 12–65 year-old vs. age ≤ 12 year-old	3.21	1.24–8.29	0.016	3.35	1.17–9.56	0.024
Age > 65 year-old vs. age ≤ 12 year-old	4.79	2.06–11.16	0.001	5.78	2.28–14.67	0.001
Female sex	1.47	0.75–2.88	0.262	0.68	0.32–1.44	0.309
Days from symptoms to diagnosis	0.99	0.94–1.06	0.971	0.98	0.93–1.05	0.591
O:4 vs. others	0.39	0.17–0.89	0.025	0.65	0.27–1.59	0.349
O:9 vs. others	1.84	0.79–4.23	0.154	2.96	1.15–7.63	0.025

aOR = adjusted Odds Ratio; CI = confidence interval.

**Table 9 pathogens-12-01298-t009:** Factors associated with antibiotic-resistant Salmonella infections: univariate and multivariate analysis.

	Univariate			Multivariate		
	aOR	95% CI	*p*-Value	aOR	95% CI	*p*-Value
Age 12–65 year-old vs. age ≤ 12 year-old	0.48	0.23–0.99	0.049	0.39	0.16–0.91	0.031
Age > 65 year-old vs. age ≤ 12 year-old	0.53	0.28–1.03	0.06	0.52	0.24–1.11	0.089
Female sex	2.12	1.17–3.84	0.014	2.94	1.47–5.87	0.002
Year of infection (per every year more)	1.37	1.13–1.67	0.002	1.50	1.19–1.89	0.001
O:4 vs. others	2.67	1.39–5.14	0.003	2.18	1.06–4.47	0.035
O:9 vs. others	0.41	0.17–0.98	0.045	0.22	0.08–0.63	0.004

aOR = adjusted Odds Ratio; CI = confidence interval.

## Data Availability

Pseudo-anonymized database will be available upon reasonable request.

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
