# Peer review of "Invasive and Non-Invasive Human Salmonellosis Cases Admitted between 2015 and 2021 in Four Suburban Hospitals in the Metropolitan Area of Milan (Italy): A Multi-Center Retrospective Study"

_pathogens, 2023, doi:10.3390/pathogens12111298_

Round 1

Reviewer 1 Report

Comments and Suggestions for Authors

Author Response

Regarding to manuscript 2636370, entitled ”Invasive and non-invasive human salmonellosis cases admitted between 2015 and 2021 in four suburban hospitals in the metropolitan area of Milan (Italy): a multi-center retrospective study”, authors reported respective study of invasive and non-invasive human salmonellosis in serogroup serotypes, clinical characteristics and antibiotic susceptibility. However, this manuscript shall be rewritten, especially in Materials and Methods, please do not write as protocol, and shorten the manuscript.

We thank the reviewer for his inputs.
The authors adhered to STROBE guidelines for the structure of this work.

Other revision is as the following:

  1. Extra-intestinal infection shall include as invasive infection, why did authors only

selected bacteremia?

Our focus was to study the epidemiological and clinical differences between bacteriemic and non-bacteriemic patients, as it’s the most common extra intestinal manifestation. Non bacteriemic invasive infections were poorly represented in our casistic (accounting for only six patients).

  1. Authors did not answer that serogroup or serotype are important for bacteremia?

    In multivariable analyses O:9 serogroup was independently associated to bacteremia (vs. other serogroups, aOR 2.96), while O:4 wasn’t. These two groups were selected based on frequency data reported in Tab 3 (frequencies of these groups were the only ones that statistically differed between bacteremic and not bacteremic).

  2. Authors shall present only in epidemical and clinical study for important serotype and

decipher the importance of these serovars to cause bacternia.

  1. Authors shall only discuss antibiotic susceptibility for these serovars.

    We appreciate the reviewer’s suggestion, and we’ll consider them for possible future publications.
    We did however edit Results and Discussion sections according to rev2’s suggestions.

Reviewer 2 Report

Comments and Suggestions for Authors

The manuscript describes the attributes of invasive and non invasive salmonella infections in Italy. Although it is well planned retrospective study, the data set is not very large to generalize the findings. There are some interesting findings which can be presented in better way for the interest of readers. 

1) Authors may include more data and enhance the scope to more than 4 years i.e 10 years to see the evolution of invasive infections and AMR in salmonella or include more centres from the neighbouring cities as well, if possible to make this more interesting study. 

2) Presentation of the results and discussion need to rewrite in concise and more attractive way. 

3) there are many grammatical errors which need corrections. 

4) more comments are in attached file

Author Response

The manuscript describes the attributes of invasive and non invasive salmonella infections in Italy. Although it is well planned retrospective study, the data set is not very large to generalize the findings. There are some interesting findings which can be presented in better way for the interest of readers. 

The authors thank the reviewer for the interest in our work.

1) Authors may include more data and enhance the scope to more than 4 years i.e 10 years to see the evolution of invasive infections and AMR in salmonella or include more centres from the neighbouring cities as well, if possible to make this more interesting study. 

As of today we don’t have the possibility to add more data to the 5 year we studied. We’ll try to address this suggestion in future works.

2) Presentation of the results and discussion need to rewrite in concise and more attractive way. 

3) there are many grammatical errors which need corrections. 

We modified the text according to reviewer’s suggestions (see below) and proof readed the article again.

4) more comments are in attached file

Notes in text were addressed and highlighted in revision mode.

Additional comments are reported below:

Lines 76-77: factors involved in S. Napoli’s increased virulence are still a matter of study.

Line 79: references for LMIC were already present, we added them at the end of the paragraph (4,5). Reference linked by rev2 is a veterinary surveillance study focused on S. pullorum.

Line 108: data were retrospectively collected from clinical records; no standard form was used. The authors feel that bullet points are more easily readable than paragraph form, if the editor agrees with rev2 we will modify it.

Lines 172-173: lines 90-92 “Our study aimed [...] to analyze the differences in clinical and epidemiological characteristics between bacteremic and non-bacteremic infections”; lines 133-134 “Body fluid on which Salmonella grew (defined as blood, feces, urine, or other)”.

Lines 232-233: resistant isolates did not significantly differ between groups. A sentence was added in the results section.

Lines 340-341: the CFR we found is very similar both to the global estimated one and to the Western Europe CFR (8.8%). As reported in Tab 1, bacteremic patients were significantly older (thus possibly with poorer health) and there was an higher frequency of immune impaired patients. While resistance is associated with mortality, it is not the only factor to account for. A sentence was added in discussion.

The comparison with pneumococcal disease is merely to put in perspective the numbers with another invasive infection known for its high lethality.

Lastly, all bacteremic patients presented themselves with a clinical picture compatible with sepsis (SOFA ≥2) so the authors think that a “subclinical” Salmonella bacteremia in patients admitted for other reasons is improbable.

Lines 361-362: we couldn’t use the “MDR” Salmonella definition because oral chloramphenicol is not available in Europe, and it is not part of EUCAST standard ABG for Salmonella (thus wasn’t tested). To meet the reviewer suggestion, we added the frequency for isolates resistant to 2 or more class of antibiotics (50/202; 24.8%).

Round 2

Reviewer 2 Report

Comments and Suggestions for Authors

much improved version

Comments on the Quality of English Language

minor edits needed. grammatical errors